# Plasticity of source-sink dynamics contributes to wheat yield stability

Tien-Cheng Wang [1], Anna Moritz [2], Mahmoud Mabrouk [1,3], Emilio Villar Alegría [1], Burak Arinalp[1], Eliyeh Ganji[4], Lukas Förter [5], Benjamin Wittkop[5], Eva Herzog [2], Rod J. Snowdon [5], Andreas Stahl[4] & Tsu-Wei Chen [1] ✉

Crop grain yield is the outcome of complex interactions among multiple physiological processes governing source accumulation via photosynthesis followed by remobilization of assimilates into grain sinks. Throughout these interdependent processes and across all developmental stages, complex genotypes by environment by management interactions have a strong impact on source and sink strength and their dynamic interactions. Recent publications proposed a conceptual "wiring diagram" of physiological traits impacting wheat yield as a framework to link quantitative genetic networks with interaction models that can help explain, predict and improve yield stability in fluctuating environments. Here we compile large-scale datasets describing historical wheat breeding progress for source-related and sink-related traits to support this concept. Furthermore, further data delivers evidence that plasticity of wheat source-sink dynamics contributes to yield stability under stress, supporting potential roles for previously unexplored traits and their interactions to maintain future yield progress in the face of climatic challenges.

The world cropping area for wheat exceeds that of any other crop, and high grain yields in intensive wheat cropping systems are essential for global food security[1]. Breeding has continuously raised wheat yields in high-input production systems[2–4], however increasing environmental and regulatory challenges for agriculture raise concerns about the potential to maintain wheat yields. A better understanding of the genetic and physiological interplay between molecular and developmental processes underlying source-sink activities and their consequence in yield formation is seen as a key to minimizing crop adaptive responses that limit yield potential[5,6]. Recently, wiring diagrams have been proposed to consider the different roles of source capacity and sink demands throughout the plant development stages, bringing the phenology and source-sink relationship into a comprehensive context[7–9]. However, source-sink relationships are impacted by the complex interplay of a multitude of environmental (E) and management factors (M). These factors further interact with the plant genotype (G), and the genotype-specific strategies to dynamically acclimatize plant genotypes to given environmental conditions[10,11], leading to higher-order G*E*M interactions, which must be considered in both breeding and agronomy[12,13]. Considering that G*E*M interactions affect all physiological processes under quantitative genetic control (for example, phenology, organogenesis, dry matter production and partitioning, photosynthetic acclimation, canopy architecture, water and nutrient uptake/transport, anthesis, senescence or grain maturation), their impact on the dynamics of source-sink relationships and yield performance is highly complex[14]. Moreover, variation in plant growth regulator signaling and sensitivity further mediates these responses[15–20], adding an additional genetically variable

¹Section of Intensive Plant Food Systems, Albrecht Daniel Thaer-Institute of Agricultural and Horticultural Sciences, Humboldt Universität zu Berlin, Berlin, Germany. ²Department of Biometry and Population Genetics, Institute of Agronomy and Plant Breeding II, Justus Liebig University, Giessen, Germany. ³Department of Agronomy, Faculty of Agriculture, Cairo University, Giza, Egypt. ⁴Julius Kuehn Institute (JKI), Federal Research Centre for Cultivated Plants, Institute for Resistance Research and Stress Tolerance, Quedlinburg, Germany. ⁵Department of Plant Breeding, IFZ Research Centre for Biosystems, Land Use and Nutrition, Justus Liebig University, Giessen, Germany. ✉e-mail: tsu-wei.chen@hu-berlin.de

layer to the regulation of plant performance. Attempts that modify only single traits have mainly failed to improve source-sink relationships[21]. Because this level of complexity is difficult to disentangle into selectable component traits, breeding of arable crops generally focuses on yield performance as the ultimate outcome of all interactions. This reliance on an aggregated end-point phenotype with comparatively low heritability makes it very challenging for breeders to effectively identify and exploit genetic diversity for adaptation of crops to new situations without compromising the yield performance. These considerations are particularly pertinent in regard to overcoming the negative impact of climate change or reduced-input cropping systems on yield and yield stability. Breeding for yield stability necessitates a comprehensive understanding of the physiological mechanisms underlying yield formation and how G*E*M interactions govern key yield-determining or yield-limiting processes.

In this study, we address the above-mentioned challenges by a data synthesis investigating the extent to which yield-based selection has indirectly influenced source-related and sink-related traits. In total, 61 traits were analyzed. Furthermore, we assessed the respective contributions of source and sink traits to historical genetic progress for grain yield. Using a German winter wheat panel comprising 202 cultivars released between 1963 and 2018, we (i) compiled datasets from 101 field experiments, incorporating comprehensive measurements, including phenology, yield components, grain quality, and disease resistance, using traditional low-throughput phenotyping protocols[22]; (ii) compiled datasets from manual measurements, model-assisted approaches, or high-throughput phenotyping in two greenhouse experiments and seven growth chamber experiments, using the same cultivar panel, to assess morphological, architectural, and physiological traits across different phenological stages from seedling to maturity (Table 1); (iii) analyzed both the absolute and relative breeding progress for all traits and assessed how trait plasticity in response to environmental conditions exhibit their breeding progress; (iv) investigated whether these traits exhibit signatures of selection; and (v) compared the stage-specific sensitivity of yield components between modern and historical cultivars[11] to environmental variables. Finally, we integrated the observed breeding progress for individual source and sink traits into an interconnected wiring diagram, based on the structure proposed by literature[7], and tested the hypothesis that breeding for plasticity in source-sink dynamics throughout the life cycle of winter wheat contributes to its yield stability.

## Results

### Data-synthesis elucidate breeding pathways that realize the concurrent improvement of source and sink traits

Our data-synthesis utilized six large datasets to analyse trait variation in central European winter wheat across different experimental setups, including field, greenhouse and growth chamber environments

(Table 1). The datasets encompass cultivars released between 1963 and 2018, with varying numbers of environments and cultivars per dataset (Supplementary Fig. 1 and Supplementary Data 1). The BRIWECS1 and BRIWECS2 datasets, described in detail by Wang et al. (2025)[22], cover the largest range of environments, while datasets WheatSouSi1, WheatSouSi2, Phenoplast and Lichthardt provide additional trait data from further field, greenhouse and growth chamber experiments, respectively (Table 1). The data sources include published studies[4,11,22–26] as well as previously unpublished datasets. The comparability of breeding progress across datasets is ensured because a large proportion of the investigated cultivars overlapped across all experiments. Specifically, BRIWECS2, WheatSouSi1, WheatSouSi2 and Phenoplast are subsets of cultivars in the BRIWECS1 dataset, supplemented by additional cultivars released in later years (Supplementary Fig. 1).

Absolute and relative breeding progress[24] was assessed for a total of 61 traits (Supplementary Fig. 2 and Supplementary Data 2), including: (1) 26 source-related traits, five of which were disease-related; (2) 16 traits related to both source and sink properties; and (3) nine sink-related traits, one of which was disease-related. In addition, ten traits covered phenology, grain quality and yield. Collectively, these datasets provide comprehensive information across the life cycle of winter wheat, from the seedling stage to maturation (Supplementary Data 2). Since source traits are traditionally considered key contributors to biomass accumulation, they can be further categorized into light interception efficiency and light use efficiency, representing different aspects in the Monteith equation framework for biomass accumulation[27,28]. In addition, disease resistances that maintain canopy function are also considered to be source-relevant traits. Eight traits related to light interception efficiency include leaf length (LL) and leaf width (LW) for single seedling and flag leaves, and their ratio (LW ratio), single seedling leaf area (LA), canopy leaf area index (LAI), light interception efficiency (LIE), tiller angle (TA) measured from the vertical axis of the canopy and light extinction coefficient (K). Thirteen traits associated with light use efficiency include traits at the leaf and canopy levels. Traits at leaf level include leaf mass per area (LMA), leaf chlorophyll concentration (SPAD), maximal chlorophyll synthetic rate (S max), chlorophyll degradation rate (Dr), the chlorophyll longevity parameter (Td), stomatal density (SD), stomatal length (SL), stomatal size (SS), stomatal width (SW) and maximal stomatal conductance estimated from morphological traits (Gs max). Traits at canopy level include light use efficiency (LUE), green canopy duration (thermal sum from heading until 50% canopy senescence, GCD) and speed of canopy senescence (GCD.S). Traits that potentially serve as an interface between source and sink were categorized as source/sink-related traits[29–32]. These include: (1) six traits related to the size and number of storage organs that depend on source supply during early developmental stages but later act as source providers (e.g., water-soluble

**Table 1 | Summary of data sources used in the data synthesis, detailing the types of experimental setups (e.g., field, greenhouse, growth chamber), the number of cultivars per dataset, and the range of release years of the studied cultivars**

| Dataset | Number of cultivars | Year of release | Number of experiments per trait | Type of experiment | Temperature (°C) | Global radiation (MJ/m²) | Source |
|---|---|---|---|---|---|---|---|
| BRIWECS1 | 191 | 1966–2013 | 3–59 | Field | 6.6 ~ 9.6 | 7.4 ~ 11.3 | 22 |
| BRIWECS2 | 52 | 1963–2016 | 1–43 | Field | 5.8 ~ 10.1 | 2.9 ~ 11.2 | 22 |
| WheatSouSi1 | 50 | 1966–2018 | 1 | Field | 11.3 | 11.8 | Unpublished |
| WheatSouSi2 | 50 | 1966–2018 | 2 | Greenhouse | 18.7 ~ 20.3 | 7.1 ~ 9.8 | Unpublished |
| Phenoplast | 50 | 1966–2018 | 3-4 | Growth chamber | 14.2 ~ 29.7 | 0.93 ~ 8.7 | Unpublished |
| Lichthardt | 191 | 1966–2013 | 1–3 | Field | 8.2 ~ 9.1 | 9.1 ~ 10.1 | 24,25 |

Ranges of temperature and global radiation in each dataset were calculated from the minimum and maximum mean values observed across experiments.
Each dataset includes measurements of various traits across diverse experiments (Here refers to combinations of years, locations, treatment), resulting in a range of environmental conditions represented. Complete lists of cultivars and traits are provided in Supplementary Data S1, S2, respectively. The photoperiodic length of experiments in WheatSouSi2 and Phenoplast is 14/10 h and 12/12 h, respectively.

carbohydrates or nitrogen stored in vegetative organs), and (2) ten traits associated with organ development that require source input but also affect light interception and, consequently, source capacity (e.g., tiller development). In our analyses, sink traits refer to traits impacting the development of the spike, which acts as the major carbon sink. These include eight key yield components such as thousand grain weight (TGW), spike number at maturity (SN), spike number at heading (SN.59), grain number per spike (GpS) and grain number (GN = SN x GpS), ratio of spike number at maturity to tiller number at heading (SN.tiller ratio), dry matter of spike at heading (DM.spike.59), and dry matter of spike at maturity (DM.spike.87). In addition, Fusarium infection affects spike development and is considered as a sink trait of spike.

To ensure the comparability between traits, we used the relative breeding progress to describe the change of the traits between 1970 and 2010 (Supplementary Fig. 2B). Over these 40 years, wheat breeding increased the source capacity by an average increase of 6% in light use efficiency, while light interception efficiency remained unchanged (2%; Fig. 1a and Supplementary Data 2). Light use efficiency (LUE) has improved due to an average increase of 1.4% in maximal chlorophyll synthesis rate and 4% in delayed leaf senescence, reflected in 5% increase in leaf chlorophyll content and 4% decrease in chlorophyll degradation rate. In addition, the increase of 2% in stomata width and 2% of stomata length leads to 5% increase in stomata size. Together with 6% increase in stomata density, the improvement of maximum gas conductance by 7%, implying that breeding progress has enhanced gas exchange and therefore light use efficiency (Fig. 1a). Analysis of stem water soluble carbohydrates (WSC) confirmed that the improvement in photosynthetic capacity consistently increased the available carbon storage in leaf and in stem at anthesis by an average of 10% and 7%, respectively. These results at the leaf and stem level reflect photosynthetic capacity. At the canopy level, a 10% increase in canopy green duration further improved the duration of source availability. Although LUE was calculated over the entire life cycle, changes in leaf mass per area, chlorophyll content, tiller number, and tiller angle in the seedling stage (Fig. 1) indirectly suggest that improvement in LUE also during early developmental stages.

The mild increase we observed in leaf area index was not reflected in improved LIE (Fig. 1a). Although light interception efficiency remained unchanged, canopy architecture has been modified by the breeders. For example, the 11% reduction in leaf size due to average reductions of 2% in leaf length and 5% in leaf width, respectively. However, an average increase of 1% in tiller numbers between 1970 and 2010 (Fig. 1b) could explain the 3% increase in leaf area index (Fig. 1a). Reducing single leaf area will normally reduce the light extinction coefficient, K[33]. However, an 4% increase in tiller angle, together with the increased tiller number, could also increase mutual shading, leading to 1% reduction of K[34]. This explains the observed lack of change in K at the canopy level. Nevertheless, this architectural change likely still contributes to a different light distribution within the canopy[35,36] and can therefore additionally contribute to an 6% increase in LUE at the canopy level[36–38].

For source-sink traits (Fig. 1b), breeding between 1970 and 2010 has increased the dry mass of stems by an average of 9% at heading. Since plant height was simultaneously reduced by an average of 10% and leaf dry mass before anthesis by 1%, this implies a potential increase in stem thickness. Spike weight, a key sink trait, has also been improved, with an increase in spike dry mass at heading for 13% and at maturity for 20% over the 40 years, primarily driven by 16% increase in grain number per spike and 3% increase in grain weight. An 2% increase in total spike number at maturity is related to a 4% increase in tiller number at heading, indicates increased sink capacity was supported by a stronger source. In addition, breeding has led to an average 4% extension in the anthesis-to-maturity period in modern cultivars, supporting delayed senescence and prolonged grain filling sustained by the simultaneously increased levels of available source WSC. Although breeding did not alter nitrogen concentrations in leaves and stems, the concurrent increase in tiller number and leaf mass per area in modern cultivars likely enhanced pre-anthesis carbon and nitrogen storage capacity, supporting grain filling. Also, breeding improved key grain quality traits, such as grain protein content (+13%) and falling number (+4%), alongside yield, indicating coordinated progress in both carbon assimilation and nitrogen acquisition[39,40].

In regard to breeding progress for fungal disease resistances (Supplementary Fig. 3), small incremental reductions in infection scores were observed for stripe rust, leaf rust and Septoria, along with a large reduction in powdery mildew, indicating some progress in resistance breeding contributing further to increased source capacity. However, a lack of improvement in resistance against Fusarium head blight and DTR demonstrates the need for further efforts in resistance breeding.

For most traits, a significant selection signal was detected in at least one of the investigated environments or developmental stages (Fig. 2). In general, the detected selection signals align with the breeding progress evaluated in Fig. 1. Interestingly, the most consistent signals across different environments were observed for complex traits such as grain yield, harvest index, plant height and protein yield, which have been direct targets of selection in the past. In contrast, selection signals for source and sink traits, which are likely subject to indirect selection, were weaker and less consistent across environments. This indicates that these less complex traits are more adaptive to environmental factors, making them unexploited, potential targets for environment-specific breeding.

## Breeding increases plasticity in adaptive physiological traits and reduces plasticity in constitutive traits

We defined trait plasticity in a given cultivar as the ratio between the range and mean of trait values across all environments. Level of plasticity varied widely between traits and genotypes (Supplementary Fig. 4). Tiller number, tiller angle, canopy/straw dry matter, leaf area index, single leaf area, leaf mass per area, stomatal size and stomatal density show high plasticity (medians of all cultivars range from 53–238%), while traits such as light use efficiency and chlorophyll content are less plastic (11-12%). This highlights the diverse potential of traits and of cultivars to adapt under changing environmental conditions.

Since the degree of plasticity is independent of the unit for each trait, absolute breeding progresses of plasticity in different traits are directly comparable. Notably, adaptive physiological traits that govern stress adaptation process but do not consistently impact yield in all environmental conditions[41] exhibited increased (leaf area, leaf width and tiller to spike ratio) or decreased (stomatal density and maximal stomatal conductance) plasticity in modern cultivars (Fig. 3). In contrast, constitutive traits that consistently contribute positively to yield across environments showed reduced plasticity as a consequence of breeding. These included light interception efficiency (LIE), light use efficiency (LUE), chlorophyll content (SPAD), light extinction coefficient (K), leaf area index (LAI), green canopy duration (GCD), spike dry matter (DM.spike.59, DM.spike.87), leaf dry matter (DM.leaf.59) and plant height (Height). Our findings provide evidence that breeding has systematically changed the plasticity of adaptive physiological traits critical for environmental adaptation, suggesting a potential "meta-mechanism" that contributes to long-term yield stability.

## Breeding increases yield resilience under climate change by improving the stage-specific sensitivity of yield components to short-term temperature fluctuations

Significant breeding progress in traits related to source-sink characters at different developmental stages (Fig. 1) may have also influenced the stage-specific sensitivity of yield components to

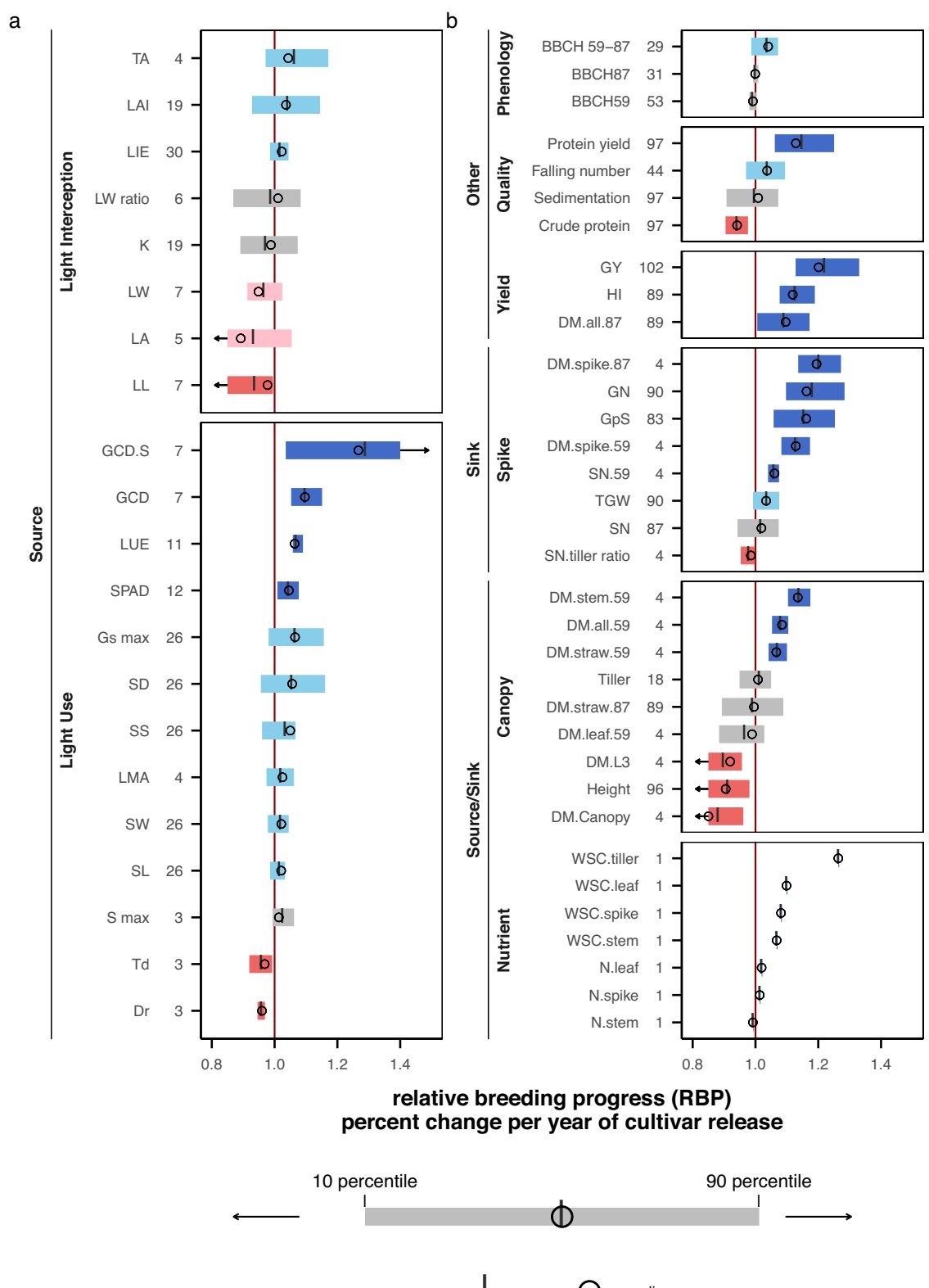

**Fig. 1 | Data-synthesis of relative breeding progress between 1970 and 2010.**
**a** 21 source traits and (**b**) 14 source-sink traits, seven sink traits, and 10 other agronomic and quality traits. Abbreviations and statistics of absolute breeding progress are detailed in Supplementary Data S2. The red vertical line ($x = 1$) represents no change of relative breeding progress over time. Numbers on the y-axis indicate the number of environments analyzed (combinations of year, location, and treatment across all experiments). Boxplots display the mean (short vertical lines), median (open circle), 10th and 90th percentiles (colored box range), with arrows indicating outliers beyond the x-axis range. Notably, the sub-category 'Nutrient' in (**b**) represents results derived from a single environment. Dark blue and red boxes signify a clear increase or decrease in trait values over time, respectively, while light blue and pink boxes indicate moderate trends. Traits with nearly no change in breeding progress (delta < 2% from x-axis value of 1.0) are shown in gray.

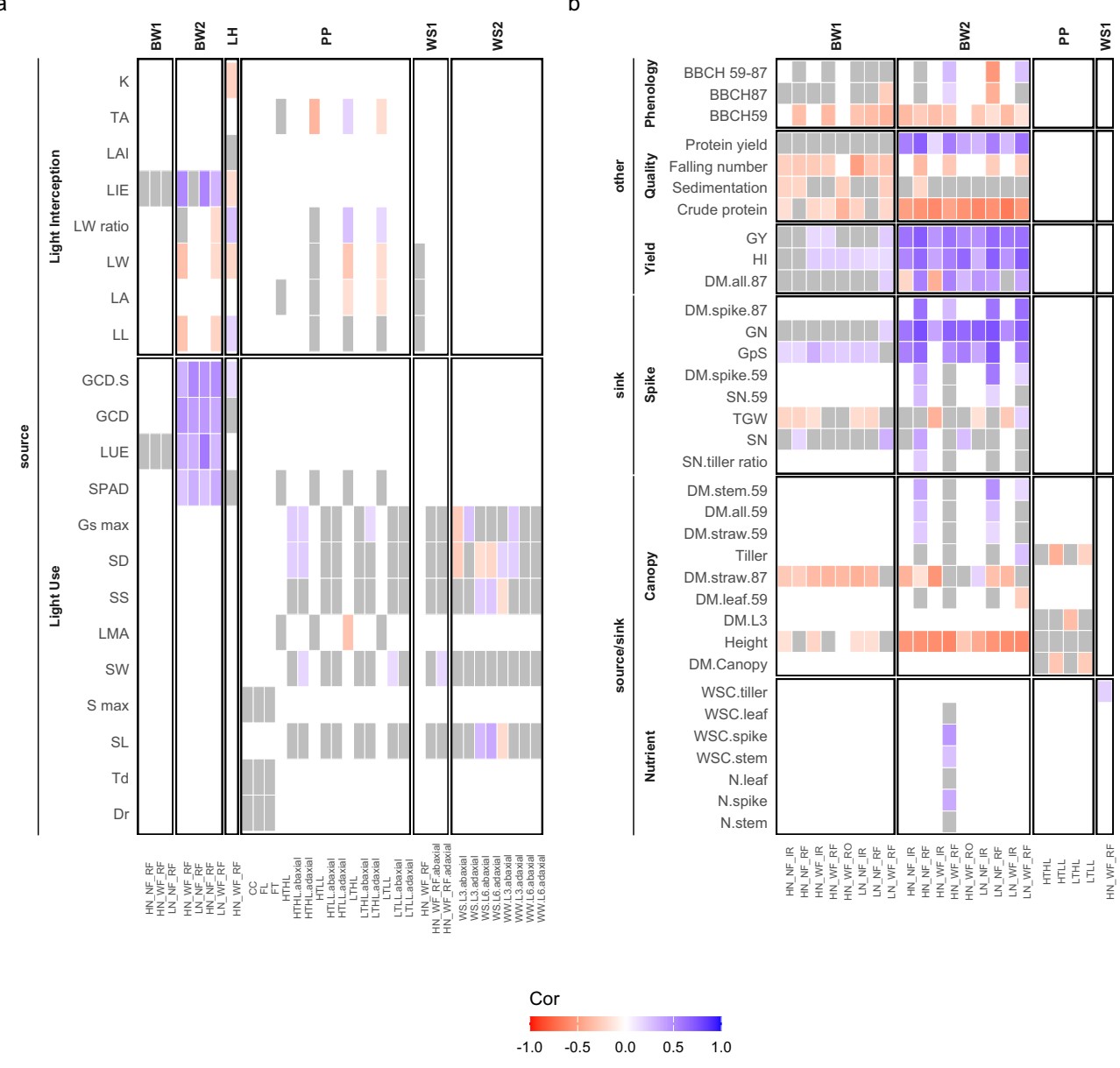

**Fig. 2 | Selection signals for source and sink traits across environments.**
**a** Source traits and (**b**) source-sink traits, sink traits, and other agronomic and quality traits. Trait abbreviations are defined in Supplementary Data S2. The x-axis represents different treatments, environments or developmental stages in which the traits were measured, with results grouped by the dataset used for calculation (BW1: BRIWECS1; BW2: BRIWECS2; LH; Lichthardt; PP: PhenoPlast; WS1: Wheat-SouSi1, WS2: WheatSouSi2). Cell color indicates the direction and magnitude of the correlation coefficient (Cor) between marker effects and allele frequency changes. Violet shading indicates a positive selection signal, while red shading represents a negative selection signal. Statistical significance was assessed by the permutation-based Ghat-test for selection[104]. Colored cells indicate significant selection signals ($p < 0.1$), whereas non-significant results are shown in gray. White areas indicate the absence of data for a given environment. Full statistical results (Ghat statistic, $p$-values and correlation coefficients) and environments are provided in Supplementary Data S3.

short-term environmental fluctuations. To investigate this, we reanalyzed a published dataset[11,42] and compared 31 old cultivars (released before 1980) with 38 modern cultivars (released after 2010). The analysis revealed that effects of global radiation, precipitation and temperature on yield components differ between old and modern cultivars, particularly during critical developmental subphases between the double-ridge stage to grain desiccation (Fig. 4). In both old and modern cultivars, grain yield was less sensitive to environmental variables than yield components, in accordance with the idea of "meta-mechanism" that stabilizes more complex traits in commercial cultivars by reducing sensitivity to short-term environmental fluctuations.

We compared the differences in short-term environmental sensitivity between old and modern cultivars (Supplementary Fig. 5). Surprisingly, despite higher and more stable yield in modern cultivars, they exhibited greater positive sensitivity than old cultivars to environmental variables. Notably, the most significant differences between old and modern cultivars lie in temperature sensitivity. In general, modern cultivars respond more favorably to increased temperatures, both before and after anthesis, highlighting the impact of breeding in adapting cultivars to rising temperatures as a consequence of climate change[43–45]. For example, higher minimum night temperatures ($T_{min}$) promoted grain yield in modern cultivars during the period between the double-ridge and terminal-spikelet stages (680–600 °Cd before

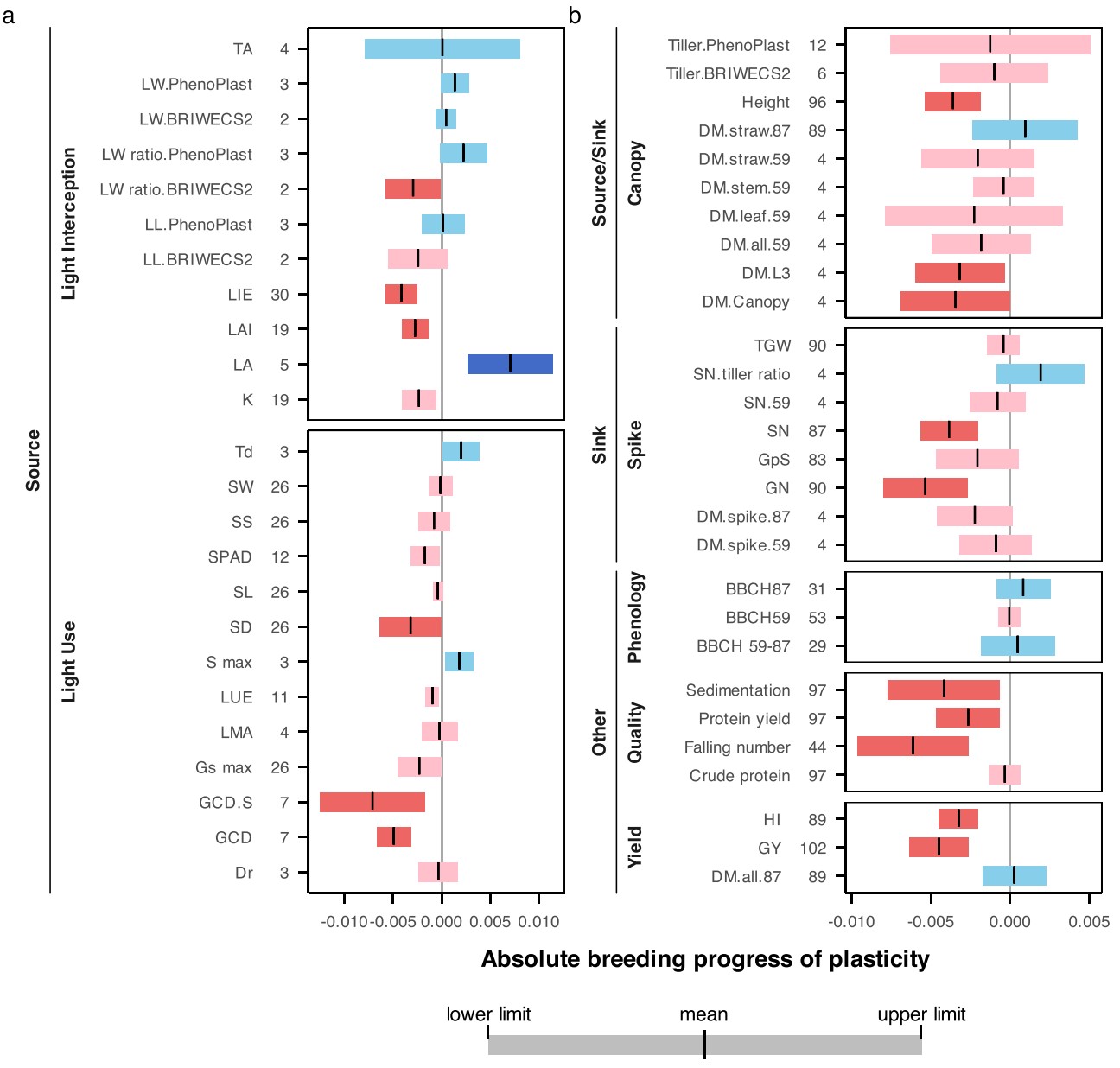

**Fig. 3 | Absolute breeding progress in trait plasticity across all environments. a** Source traits and (**b**) source-sink traits, sink traits, and other agronomic and quality traits. negative values. Trait abbreviations are defined in Supplementary Data S2 and statistical details are shown in Supplementary Fig. 4. The short vertical lines in the boxplot represent the estimated absolute breeding progress, and the boundary represent the confidence interval. Data of LL, LW, LW ratio, tiller number obtained from field experiments or control environments are calculated separately to avoid unbalanced numbers of cultivars between datasets.

heading, Fig. 4). During this developmental window, $T_{min}$ ranged from $0.21 \pm 0.43\,°C$ to $11 \pm 0.69\,°C$ across the studied cultivars. Between pre-grain filling and grain filling stages (260 and 300 °Cd after heading), higher $T_{min}$ promoted grain per spike (GpS) in modern cultivars more (Fig. 4d) than in old cultivars (Fig. 4b), in the range between $8.4 \pm 0.41\,°C$ and $17.8 \pm 0.58\,°C$.

Direct comparisons between old and modern cultivars during the same time window are challenging due to differences in phenology in the studied panel[4,24], hence results were carefully interpreted within this context. For example, we observed a significant positive effect of increased $T_{min}$ on thousand grain weight (TGW) during early spike development (double ridge to terminal spikelet development, 700-580 °Cd before heading) in both old and modern cultivars (Fig. 4a, b, where $T_{min}$ ranged from $0.1 \pm 0.5\,°C$ to $11 \pm 0.85\,°C$). This suggests that higher night temperatures during this period enhance TGW. However, the timing of this effect differed: In old cultivars, $T_{min}$ had the strongest

impact between −700 and −620 °Cd, whereas in modern cultivars, the effect was most pronounced between −660 and −580 °Cd. Assuming the physiological stages of this response align, this would indicate a developmental shift of approximately 40 °Cd, with modern cultivars exhibiting a shorter duration from this stage until heading. Indeed, this expectation is consistent with our previous observations[4,22,24,46]. Interestingly, modern cultivars exhibited a stronger positive response of TGW to $T_{min}$ (8.57%, Fig. 4c) than older cultivars (6.16%, Fig. 4a), reflecting the consistent $T_{min}$ effect on overall yield.

Old and modern cultivars exhibited different sensitivities to mean temperatures ($T_{mean}$) and maximum temperatures ($T_{max}$), which occurred during the phase between the double-ridge stage and terminal-spikelet initiation (between approximately 600 and 450 °Cd before heading; where $T_{mean}$ ranged from $5 \pm 0.2\,°C$; to $18.2 \pm 1.5\,°C$ and $T_{max}$ ranged from $8.7 \pm 0.16\,°C$ to $26.3 \pm 1.8\,°C$; Fig. 4). This indicates that 1) selection under climate change has also reduced the

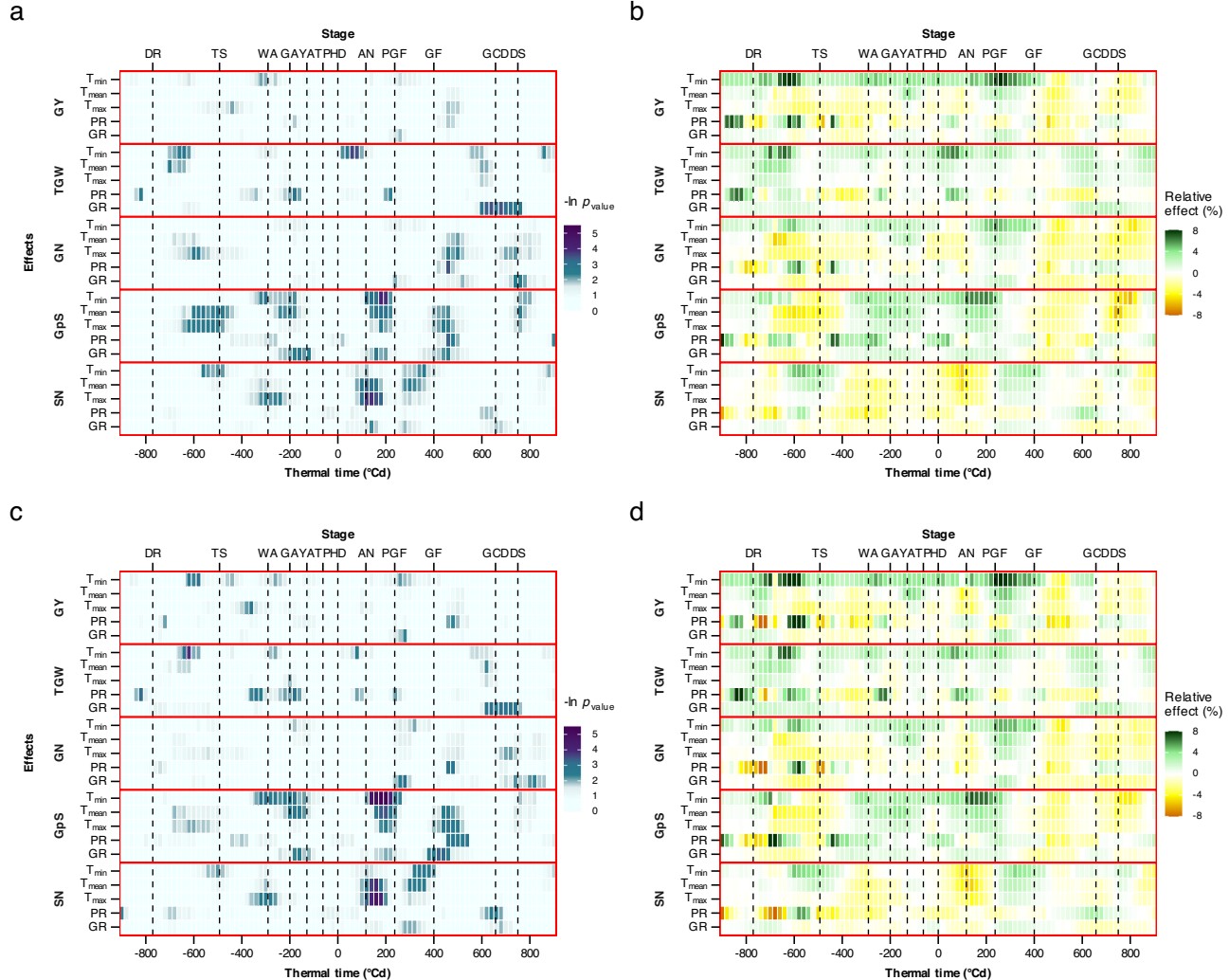

**Fig. 4 | Sensitivity of yield and yield components (GpS, SN, GN and TGW) to short-term environmental variables (GR, $T_{max}$, $T_{mean}$, $T_{min}$ and PR) before and after heading (thermal time = 0 °Cd). a, b** Old cultivars released before 1980 ($n = 31$). C-D: modern cultivars released after 2010 ($n = 38$). **a, c** Median significance of ANOVA −ln(*P*-value) of selected cultivars. Significance assessed through *p*-value from a two-sided *t*-test of analysis of variance. **b, d** Median sensitivity (normalized coefficient of regression, in %) of yield components to per unit change of environmental variables. Positive and negative effects are shown in green and yellow, respectively. Details are presented in the published data[11,42]. Yield and different yield components are separated by solid red lines. AN, anthesis; DR, double-ridge stage; DS, desiccation; GA, green anther stage; GCD, green canopy duration characterized by 50% canopy senescence; GF, grain filling; HD, heading stage; PGF, pre-grain filling; TS, terminal spikelet stage; WA, white anther stage; YA, yellow anther stage.

negative response of grain number per spike to rising temperature and 2) modern cultivars exhibited enhance sink resilience to short-term spring heat, probably driven by improved source capacity supporting faster sink organ development under higher temperatures at this stage. Also, compared with old cultivars, the grain number per spike in modern cultivars showed a stronger increase when $T_{min}$ (ranged between $5.3 \pm 0.43$ °C and $14.4 \pm 0.43$ °C) increased between green anther to yellow anther stage (180 and 140 °Cd before heading), and $T_{mean}$ (ranged from $12.5 \pm 0.21$ °C to $22 \pm 0.34$ °C) between anthesis to pre-grain filling stage (140 – 260 °Cd after heading).

Furthermore, between the pre-grain filling and grain filling stages (320–340 °Cd after heading), spike number was positively correlated with increased day and night temperature ($T_{mean}$ ranged from $13.7 \pm 0.43$ °C to $27.1 \pm 1.4$ °C and $T_{min}$ ranged from $9.3 \pm 0.56$ °C to $18.6 \pm 0.93$ °C, respectively). A similar response was seen in regard to global radiation. Consistently, this stage-specific effect improved the grain number and therefore yield, especially in the modern cultivars.

## Discussion

We assessed 61 morphological, architectural, agronomic and physiological traits in winter wheat across various phenological stages, from seedling to maturity, using manual measurements, model-assisted approaches and high-throughput phenotyping. Our data synthesis integrates comprehensive data from multi-environmental field trials[11,22,24], along with greenhouse and growth chamber experiments. This enabled us to dissect key eco-physiological and genetic aspects of wheat improvement through breeding over the past 60 years, interpret the results in the context of source-sink dynamics using a wiring diagram and provide insights into "meta-mechanisms" underlying improvements in yield stability[7–9].

Over the past 60 years, breeding has systematically improved source–sink traits across different developmental stage (Fig. 5). These improvements led to the emergence of "meta-mechanisms" in commercial cultivars—integrated physiological strategies that stabilize complex traits such as grain yield by reducing their sensitivity to short-term environmental fluctuations (Fig. 4). Here we discuss factors

contributing to the improved meta-mechanism in the modern cultivars. First, revisiting the breeding history of wheat reveals key agronomic and physiological changes that have shaped the phenotypic plasticity of modern cultivars in response to environmental variability. Notably, breeding has enhanced plasticity in several adaptive physiological traits[41], which underpin the plant's ability to cope with unpredictable environmental fluctuations (Fig. 3). In particular, this involved reduced plasticity in traits that support constitutive improvements in both source-related functions (such as light interception and light use efficiency) and sink-related traits (like spike number and plant height) across diverse environments. This indicates that, in modern cultivars, disturbances to either source or sink organs during sensitive developmental phases may be compensated by traits expressed under more favorable conditions in later stages, consequently maintaining overall yield. This functional flexibility across phenological stages, where the identity and demand of source and sink organs change dynamically according to the wiring diagram (Fig. 5), suggests that yield stability arises from integrated trait networks that balance resource allocation, which may involve the plant growth regulators[47,48]. In addition, a reduction in both the mean and plasticity of plant height and a simultaneous reduction in plasticity of leaf area index likely reflect selection for less competitive and more cooperative canopy structures, promoting collective productivity among neighboring plants[49]. Furthermore, the phenotypic plasticity of traits that mediate source−sink coordination emerges as a key target for buffering yield against environmental stress. For example, while the plasticity of tiller number has decreased, plasticity of the spike number-to-tiller number ratio has increased, indicating a shift towards optimizing reproductive efficiency under variable conditions. Moreover, we emphasize that yield stabilization can be governed by different physiological meta-mechanisms. Detailed studies of 130 traits across the life cycle of eight cultivars—also included in our analysis—revealed that modern cultivars employ at least two distinct physiological pathways to stabilize yield[50].

The selection of modern cultivars took place under changing climatic conditions, which has likely contributed to their improved responsiveness to night temperatures before the grain filling stage (Fig. 4). This suggests that breeding has played a role in mitigating the negative impacts of rising temperatures associated with climate change[13]. We did not observe negative effects of elevated night temperatures on grain yield and other yield components around anthesis or grain filling, as reported in a previous meta-analysis[51]. This is probably because the experimental setup in that analysis used prolonged increased temperature treatment, whereas we looked at short time windows of just 2-3 days. In our experiments, modern cultivars benefitted from higher minimum night temperatures by showing increased thousand grain weight (TGW) during early spike development (700 to 580°Cd before heading, where night temperatures normally remained below 11 °C, Fig. 4). They also showed enhanced grain number per spike during both the green to yellow anther stage (180–140 °Cd before heading) and the anthesis to pre−grain filling stage (140–260 °Cd after heading; where mean temperature was normally below 22 °C, Fig. 4). Although the mechanisms underlying these improvements remain unclear, night temperatures may influence pollen fertility, whole-plant carbon balance[52,53] and canopy water use[54,55] − factors that merit further investigation[56]. This aligns with recent findings indicating that modern cultivars show improved acclimation of photosynthetic capacity to nocturnal warming[57].

Because yield reflects the dynamic balance between source and sink[58−60], wheat breeders would not have been successful had their selection focused solely on single traits, such as canopy greenness (source capacity) or grain number (sink strength). Indeed, an unbalanced improvement of either source or sink traits cannot, on its own, ensure proportional gains in grain yield unless the complementary trait is enhanced in parallel. For example, during the critical period around anthesis, grain/sink development is highly sensitive to source

limitations caused by environmental fluctuations, such as reduced radiation[11,61,62]. Without the improvement in water-soluble carbohydrate reserves in the straw at anthesis (Fig. 1b), modern cultivars would be less able to compensate for short-term reductions in source availability and sustain grain development during this vulnerable phase. This underscores the importance of enhancing canopy sink capacity to accumulate pre-anthesis carbon reserves that can be flexibly mobilized as a source, particularly under stress conditions[63,64]. However, even in the absence of detailed physiological knowledge, empirical selection for complex traits like yield across diverse environments has likely driven the selection of meta-mechanisms that stabilize yield in modern cultivars. This may explain reported evidence for the co-evolution of source and sink traits during breeding[24] and the distinct physiological strategies by which modern cultivars achieve high yield[22]. To strategically realize the co-improvement of source and sink traits for yield stability, it is vital to revisit the genetic structures underlying grain yield related trait-trait correlation or trait-trait-coupling from a source-sink perspective[65]. Since grain yield is governed by quantitative trait loci (QTL) with pleiotropic effects, traits beyond yield itself will often be co-selected. For instance, the concentration of stem water-soluble carbohydrates, a source trait, is negatively associated with root development under drought stress[66,67] and with tiller growth, but positively linked to grain weight and grain yield[68]. We found a simultaneous increase in stem dry matter at maturity and reduction in plant height, implying a potential increase in stem thickness[69] which may be linked with the increase in stem WSC storage (Fig. 1). Together, these trait-trait linkages highlight the intricate interplay between carbon, nitrogen and water availability across canopy and root systems, ultimately influencing pre-anthesis carbon reserves and source-sink dynamics. The strong genetic linkage between above-ground and below-ground resource acquisition and photosynthate allocation necessitates an integrated approach that incorporates root traits into breeding strategies aimed at improving water use efficiency[70,71]. Furthermore, although we do not explicitly focus on biotic stressors, improved resistance to fungal diseases helps maintain functional green leaf area and thereby directly affects source capacity and associated sink processes. We identify traits requiring attention in the future to achieve dynamic source-sink balances. A significant knowledge gap remains in the precise quantification of sink-related traits, such as leaf initiation rate, tiller development, floral development, spikelet number, and spikelet size. In particular, the energy cost associated with organ development − especially the spike development occurring between stem elongation and booting − reflects the absolute strength of sink activity during this period. However, empirical data on these processes are limited, although our stage-specific analyses (Fig. 5) indirectly indicate that breeding efforts have already improved the source-sink balance within this developmental window and increased environmental resilience of spike development. The spike dry weight at heading recorded in our dataset (DM.spike.59; Fig. 1) represents an integrated outcome of the preceding source−sink dynamics, rather than a direct measure of sink strength during development. Consequently, assessing the genetic potential to optimize source−sink balance during these critical phases remains an important future research priority. Nonetheless, the quantification of sink-related traits, as well as measurements of nitrogen and water-soluble carbohydrates in developing organs across sub-phases, requires destructive sampling and labor-intensive methods. This methodological bottleneck limits the pace of improvement for these parameters[72−74], yet they remains crucial for understanding the mechanisms of sink organ formation and the reduction of grain number under environmental fluctuation[32,75−78]. Our results indicate improved coordination between canopy stay-green and grain filling, exemplified by a significant reduction in chlorophyll degradation rate and delayed leaf senescence in modern cultivars which enables the plant a longer period of assimilates production. Accordingly, post-

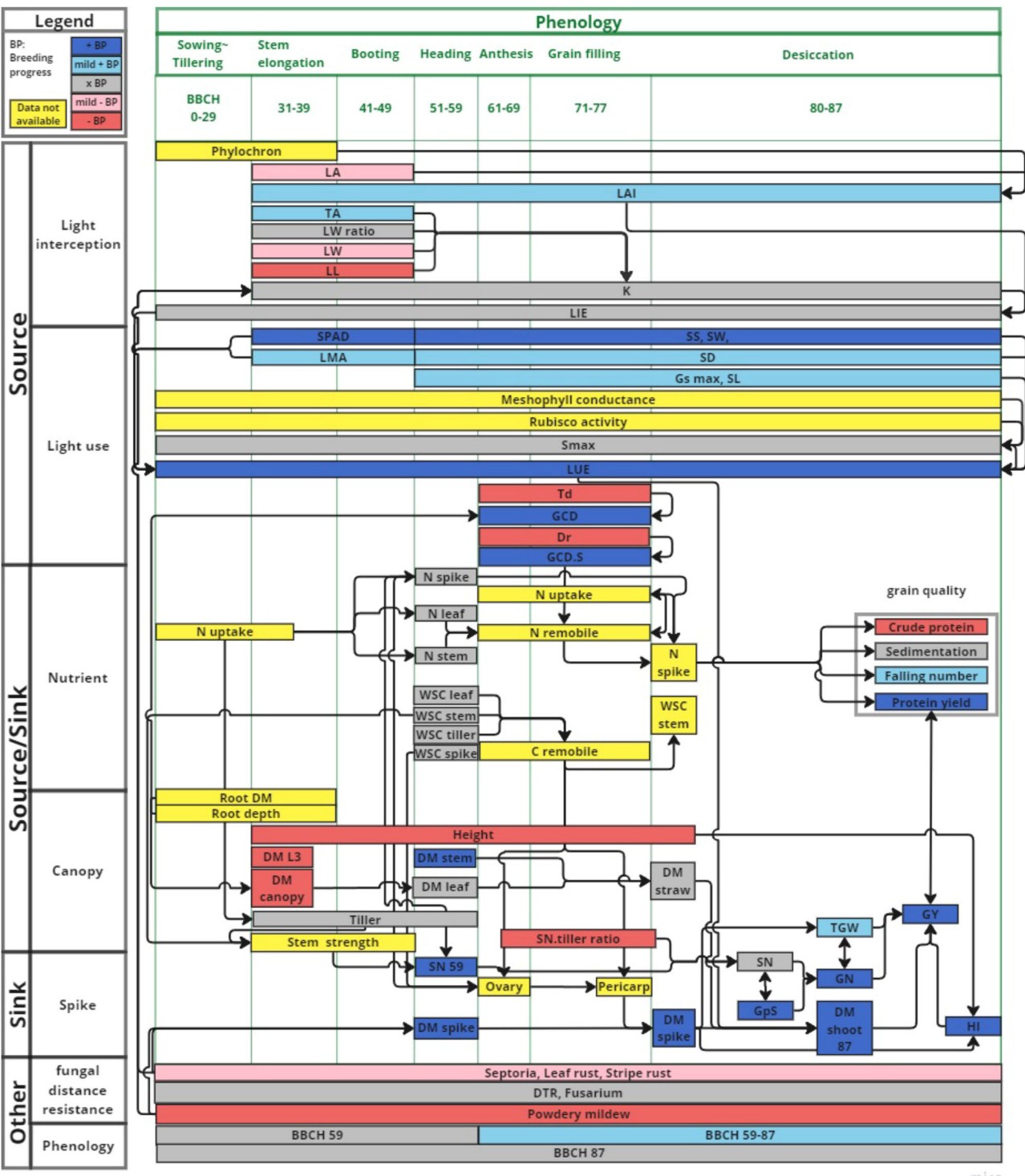

**Fig. 5 | Schematic representation of breeding progress within the stage-specific context of source-sink balance in a wiring diagram of yield formation.** The boxes at the top and left indicate the phenological stages and the different categories of traits, respectively. Each colored rectangle represents a trait, with the fill color indicating the direction of breeding progress: blue for positive progress, gray for no change, red for negative progress, and yellow for missing data. Arrows represent physiological connections between traits. Vertically, traits are grouped into source, sink, or source–sink categories. Horizontally, green frames outline the crop cycle, subdivided into growth subphases (BBCH scale) increasing from left to right. The position of each colored rectangle across these subphases indicates the growth period during which the trait exerts its impact. Traits placed outside the green columns are not confined to a specific growth stage. Usually, the increment of breeding progress represents improvement in breeding, while for some traits like plant height, Dr, Td and infection areas of fungal diseases, a reduction in the value corresponds to improvement by breeding.

anthesis source strength, maintained through stay green traits, also plays a crucial role in stabilizing yield under stress[79,80]. This highlights the need to further explore strategies for enhancing post-anthesis source strength through foliar[81] and non-foliar organs[82]. Until harvest, however, it is desirable that the largest possible proportion of nitrogen in leaves and stems is transported to the sink in order to increase the nitrogen harvest index, reduce nitrogen surplus and thereby diminish nitrogen losses to neighboring ecosystems. Ongoing improvement of yield stability in wheat cultivars could be facilitated by selection for genotypes that generate sufficient pre- and post-anthesis photo-assimilates for grain filling, in conjunction with improved sink capacity. At this point it becomes obvious that an adapted fine-tuning of the senescence is of great importance to enable an optimal source-sink balance.

Combining modeling approaches[83] for assessment of source-sink dynamics may provide avenues for predicting environmental impacts on sink organ development, dissecting the complex trait relationships and selecting for better-adapted genotypes. Breeding progress for physiological responses to temperature remains challenging to quantify and strategically elusive. High-throughput phenotyping using UAV data offers a powerful approach to quantify the nonlinear response of stem elongation to temperature[84]. However, its effectiveness depends heavily on field conditions during stem elongation, which are normally relatively mild for winter wheat, making it difficult to observe developmental barriers under extreme heat. Measuring germination speed alongside automated detection of germination stages[85] may provide a faster method for assessing temperature-driven developmental responses. However, the extent to which these responses align with the temperature sensitivity of physiological processes remains a topic of ongoing debate[86]. Modern phenotyping approaches like spectral reflectance enable high-resolution tracking of senescence dynamics and the capture of genotype-specific environmental responses across the entire crop lifecycle. Improved approaches to model complex relationships using such data, including machine learning techniques, hold considerable potential to improve our physiological understanding of multi-trait relationships and their genetic architecture.

A limitation of our synthesis that integrates six datasets across traits, environments, and management conditions is the unbalanced number of genotypes and environments represented for each trait, which is a common challenge inherent in data synthesis and multi-environment trials[87,88] and can potentially introduce weighting bias. Consequently, any cross-trait comparison of absolute effect sizes should be interpreted with caution. To mitigate this, subsets of cultivars were carefully selected before experiments to ensure the representativeness of each panel (see Supplementary Methods). When comparing plasticity, they are also estimated separately (for example, tiller number and leaf morphology; Fig. 3). Importantly, the primary aim of this data synthesis was not the direct comparison of individual trials, but rather the identification of overarching, robust trends across the wide range of environments and managements in our entire data collection. Achieving representative results and maximizing statistical power necessitated the inclusion of datasets with sufficient variation in GxExM combinations[14,87,89], even if some were structurally unbalanced, rather than restricting the analysis to a limited, balanced subset[90]. Accordingly, our datasets encompass a broad spectrum of environments—including multi-environment field trials, greenhouse experiments, and high-throughput phenotyping platforms—and multiple biological scales, ranging from leaf-level morphological traits to canopy-level leaf area index and ultimately grain yield.

## Methods
In total, data from 101 field experiments, 2 greenhouse experiments and 7 growth chamber experiments were compiled, with 206,204, 2400, 4880 genotypic values, respectively.

### Dataset collections of BRIWECS 1, BRIWECS 2 and Lichthardt
These are two published multi-environment field trials datasets including 201 winter wheat cultivars released between 1963 and 2018, evaluated under 100 experiments across six locations in Germany from 2015 to 2020[22]. In BRIWECS1, only 191 cultivars that were commonly grown in Germany and other European countries were used to calculate the breeding progress[4]. BRIWECS2 only studied 44 cultivars in the panel of BRIWECS1, with an additional eight cultivars released between 2008 and 2016[22]. Experiments were conducted using a randomized block design (2015–2017) and a full treatment-factorial design (2018–2020), with management scenarios included varying nitrogen levels (110 vs. 220 kg N ha⁻¹), fungicide application (with vs. without), and water availability (rain-fed, irrigated, or rainout-shelter). The dataset comprises 526,751 observations for 24 agronomic and physiological traits, assessed through both whole-plot and 50 cm cut sampling. Traits include dry matter (DM), grain yield (GY), thousand grain weight (TGW), spike number (SN), plant height, and grain per spike (GpS), alongside phenology and disease resistance measurements. In addition, site-specific manual measurements were conducted, including leaf morphology, canopy architecture and chlorophyll content of the leaves[25]. These were combined with process-based models to derive canopy parameters, e.g., light use efficiency, light interception efficiency, green canopy duration[24]. For details, see supplementary methods. The six fungi disease traits were measured as infection area score (%) during growth period[26]. Lichthardt dataset is collected in addition to the BRIWECS1 field experiment in Hannover, including nine source traits: leaf area index (LAI), light interception efficiency (LIE), light extinction coefficient (K), leaf length (LL), leaf width (LW), leaf length to width ratio (LW ratio), leaf chlorophyll content (SPAD), green canopy duration (GCD), speed of canopy senescence (GCD.S). The details of these nine traits is available in material and methods[24]. In addition, LIE and light use efficiency (LUE) is measured in addition to the BRIWECS1 field experiment in Kiel[4,23].

In dataset Lichthardt, which includes leaf morphology (leaf length, leaf width and leaf length to width ratio), canopy architecture (light interception efficiency) and chlorophyll content of the leaves and canopy (green canopy duration (GCD) and slope of GCD)[25]. These were combined with process-based models to derive canopy parameters, e.g., leaf area index, light interception efficiency, green canopy duration[24]. Seven dry matter traits from three organs (leaf, stem, spike) at anthesis (BBCH59) and maturity (BBCH87) (e.g., DM.leaf.59, DM.stem.59, DM.straw.59, DM.straw87, DM.spike.59. DM.spike.87, DM.all.87). Light use efficiency was calculated by the DM.all.87 divided by the total intercepted irradiance throughout the growing period in location Kiel[23]. Tiller number and spike number at heading was counted per 50 cm row cut at Hannover in 2018 (BRIWECS2).

Six nutrient traits were analyzed from grounded dry matter samples collected at BBCH59 for nitrogen and water-soluble carbohydrate analysis (e.g., N.leaf, N.spike, N.stem, WSC.leaf, WSC.spike, WSC.stem). For nitrogen and water-soluble carbohydrate (WSC) analysis, plant biomass samples from the oven were first ground into powder with laboratory knife mill Grindomix GM200 (Retsch GmbH, Haan, Germany) at 10,000 spin for 1.5 min. An anthrone based colorimetric method[91] was used for the determination of WSC, which mainly detects glucose, fructose, disaccharides, and fructosans. The main three steps were listed as followed: (1) 0.5 g of lyophilized sample powder was added with 100 ml of deionized water and shaken for 1 h; (2) The sample extract was clarified with Carrez mixture, filtered and diluted with mercury (II) chloride solution (0.01%); (3) Anthrone reactant (CFA, Scan + +, Skalar Analytical, Breda, The Netherlands) was added to determine WSC concentration using a continuous flow analyzer (CFA, Scan + +, Skalar Analytical, Breda, The Netherlands).

The nitrogen content in powdery samples was determined by elemental analysis using the Vario Max Cube (Elementar

Analysensysteme GmbH). The steps involved in the determination of nitrogen content by combustion method according to Dumas were described follows, based on[92]: (1) The sample weight ranged from 50 to 300 mg, depending on the sample material available; (2) The nitrogen bound in the sample burns under oxygen supply to molecular nitrogen (N2) and mixture of nitrogen oxides (NOx), which is subsequently also reduced to N2; (3) Argon was used as the carrier gas, the combustion temperature was 900 °C and the concentration was determined by means of thermal conductivity detector.

### Selection of Cultivar Subsets for WheatSouSi1, WheatSouSi2, and PhenoPlast Datasets

Phenology plays a crucial role in source-sink relationships and has been a major target in breeding[93–96]. To minimize potential confounding effects of phenology on our analyses of breeding progress in morphological and physiological traits, we selected a panel of 50 cultivars with similar phenology based on results from BRIWECS1 and BRIWECS2. The selection followed a three-step approach: (i) Incorporating recent breeding progress – Three modern cultivars, RGT-Reform (2014), Nordkap (2016), and Asory (2018), were included to represent breeding advancements over the last decade. In addition, Impression (2005) and Julius (2008) were pre-selected due to their contrasting canopy characteristics observed in preliminary experiments[49]. (ii) Ensuring phenological homogeneity – Only cultivars with similar heading dates in field trials[24] were selected, minimizing confounding effects from phenological variation. (iii) Maximizing genetic diversity – Genome-wide SNP profiles were used to classify the entire panel into 45 genetic clusters. From each cluster, the highest-yielding cultivar in field trials was selected to ensure representativeness. This approach effectively captures the full scope of breeding history while balancing temporal breeding progress, comparable absolute and relative yield improvements in line with the BRIWECS1 dataset, and genetic diversity, resulting in a representative subset for analyzing breeding trends.

### PhenoPlast

To examine genotypic variation in plastic responses to light and temperature, 50 cultivars were randomized in four growth chamber experiments combining two light levels (high light (HL): $458 \pm 41\,\mu\text{mol}\,\text{m}^{-2}\text{s}^{-1}$ and low light (LL): $49 \pm 3\,\mu\text{mol}\,\text{m}^{-2}\text{s}^{-1}$) with two temperature regimes (high temperature (HT): 31.29/28.04 °C and low temperature (LT): 15.86/12.6 °C day/night). Non-destructive measurements of the number of tillers were recorded weekly in the 3rd, 4th and 6th week after transplantation for three weeks prior to harvest to further investigate the plasticity of tillers development. Plant height was measured from the first internode base to the highest leaf tip. Tiller angle was estimated with image analysis software (ImageJ[97]) based on images acquired from digital camera DSC-H300 (Sony, Japan). Leaf width, length and area were measured with a handheld laser leaf area meter CI-203 (CID Bio-Science, USA). Leaf mass per area (LMA) was estimated by dividing leaf dry matter by total leaf area. SPAD was estimated from the chlorophyll meter SPAD-502Plus (Konica Minolta, Japan). Destructive measurement was carried out for the dry matter of the 3rd leaf and the shoot six weeks after transplantation. Plant samples were oven-dry at 60 °C for three days before weighting for shoot dry mass.

Morphological traits of stomata were estimated by more than 30,000 images taken by a portable high-throughput microscope (ProScope HR5, Bodelin Technologies, USA) with a 400× magnification lens[98]. Images were taken from abaxial and adaxial leaf surfaces and processed using an automatic object detection pipeline to derive stomatal length, width, size and density. For details, see the section Stomatal Morphology.

SPAD was measured twice weekly on three third leaves of 50 winter wheat cultivars from emergence to senescence (~700 °Cd).

Day/night conditions averaged 22/14 °C, 50/70% relative humidity, and 460 Photosynthetic Photon Flux Density (PPFD), with hourly fluctuations of ± 2.4 °C and ± 96 PPFD. SPAD values were converted to chlorophyll concentration ($\mu\text{mol/m}^2$)[99] and used to parameterize a model of chlorophyll turnover, providing insights into cultivar-specific chlorophyll dynamics through synthesis (S max), aging (Td), and degradation (Dr).

### WheatSouSi1 and WheatSouSi2 data collection

Fifty winter wheat genotypes were evaluated in field trials in Gross Gerau, Germany in the growing seasons 2022/2023 (WSC measurements) and 2023/2024 (stomata traits on flag leaves) in two replications following an alpha-lattice design. In each plot, above-ground plant samples of main and side tillers were taken at anthesis. The samples were then oven-dried at 50 °C for 48 h and underwent laboratory analysis for water-soluble carbohydrates. At full maturity in each plot, one meter was cut to estimate yield components, while grain yield was estimated by harvesting the whole plot.

For analysis in the laboratory, samples were ground to 0.5 mm and oven-dried to weight constancy at 65 °C. Three technical replicates per sample were taken to perform classical chemical analysis of water-soluble carbohydrates. For each replicate 50 mg of plant material was used to extract the water-soluble carbohydrates after van Herwaarden et al.[100]. Carbohydrate concentrations in the extraction solutions were estimated using the anthrone method[101]. For this D(+)-Glucose was used as a standard and extinction was measured at 625 nm.

### Stomatal morphology

An image collection of over 25,120 images of leaf microscopic morphologies was collected using a portable handheld high-throughput microscope (ProScope HR5, Bodelin Technologies, USA) equipped with 400x magnification lens, targeting different leaf ranks from field and control environments: the 3rd leaf in PhenoPlast, the 3rd and 6th leaves in Wheatsousi2, and five flag leaves per genotype in Wheatsousi1. In all environments, eight technical replicates per leaf were obtained by imaging four equidistant positions intervals on both abaxial and adaxial leaf surfaces, starting from the leaf center to the tip, using two biological replicates.

Five traits related to stomatal morphology were derived from the automatic image object detection. An object detection pipeline for stomatal morphology recognition was developed based on the YOLO v11 model (https://github.com/rayneuro/BerlinHUYoloStomata). The model was trained with 442 images for five object classes: complete, incomplete, blurry complete, blurry incomplete, or trichome. Accuracy of object detection was validated using another 240 independent images, achieving a mean average precision (mAP) of 0.77 at an intersection over union (IoU) threshold of 0.5.

Stomatal size (SS) is defined by the multiplication of stomata length (SL, long side) by stomata width (SW, short side) from an object detected with class "complete". Stomata density is defined as the total number of effective stomata counts per image area (0.806 mm²). Effective counts of stomata is a weighted sum of assigning 1 to "complete" stomata objects and 0.5 to "incomplete" stomata objects. Maximum stomatal conductance ($g_{max}$; mol m$^{-2}$ s$^{-1}$) was calculated based on stomatal density and dimensions following[102] (Eq 1).

$$g_{max} = \frac{\frac{dw}{v} \cdot SD \cdot pa_{max}}{pd + \frac{\pi}{2}\sqrt{pa_{max}/\pi}}$$

Where dw is the diffusivity of water in air (0.0000249 m² s$^{-1}$), and v is the molar volume of air (0.0224 m³ mol$^{-1}$) at 25 °C. Pore depth (pd, m) was assumed to be equal to stomatal width. The mean maximum stomatal pore area ($pa_{max}$, m²) was estimated by the formula of an ellipse, with the assumption that the major axis as pore length (i.e., half

of the half of the stomatal length) and the minor axis as half the pore length[103].

## Statistics and reproducibility
All the following analyses were conducted in the R environment (v.4.4.1).

## Estimating breeding progress and phenotypic plasticity
For each measured trait value, the best linear unbiased estimator (BLUE) was calculated for each cultivar within each experimental environment (defined as a combination of data source, location, treatment, and experimental year). Using the BLUEs, absolute breeding progress (ABP, change per year) and relative breeding progress (RBP, performance ratio of cultivars released in 2010 vs. 1970) were determined[24]. Briefly, ABP was calculated as the slope from a linear regression model, where the response was the cultivar-specific BLUE and the explanatory variable was the year of cultivar release. ABP quantifies the annual rate of change in the trait. RBP was calculated by dividing the predicted value for the 2010 release year by that for 1970, based on the linear model from ABP estimation. RBP expressed the proportional change in trait value between cultivars released in 2010 and those first released in 1970 and enables direct comparison across traits with varying units or scales (Supplementary Fig. S2).

Phenotypic plasticity for each cultivar was defined as the range of BLUEs (differences between maximum and minimum across all measured experimental environments) divided by the mean of BLUEs across all available experimental environments. This statistic estimates the extent of genotype by environment interaction (GxE) for a cultivar. A higher value reflects greater sensitivity (plasticity) of a trait to varying environmental conditions. For traits in which different experimental environments contained an unbalanced number of cultivars, calculations were performed separately for field trials and controlled environments (e.g., greenhouses or growth chambers). This approach minimized confounding effects arising from unequal cultivar representation across data sources when estimating plasticity. For leaf length, leaf width, their ratio, and tiller number, non-overlapping cultivar sets across datasets required independent calculation of BLUEs and all derived metrics. The number of cultivars included and the corresponding data sources, along with the degree of alignment across environments, are summarized in Supplementary Data S2. Because plasticity is unitless, absolute breeding progress in plasticity is directly comparable across traits and was therefore used to assess the progression of phenotypic plasticity in released cultivars over time.

## Selection signal
For each investigated trait, we determined a selection signal using the Ghat test (R package *Ghat*, v0.2.0[104]), specifically designed for quantitative traits[104,105]. To achieve this, allele frequencies were calculated based on the year of release of the varieties. The allele frequency change from the first to the last year of release was calculated for each SNP. Allelic effects were estimated with the R package *rrBLUP* (v.4.6.3)[106]. The specific genotype set used for each trait is shown in Supplementary Fig. 1. To ensure robustness, we conducted the analysis across all investigated treatments, utilizing BLUEs across multiple years and locations.

## Stage-specific sensitivity of yield components to environmental variables
We compared the differences in sensitivity of yield and yield components (thousand grain weight, grain number, spike number, and grain per spike) to short-term environmental variables between 31 old cultivars (released before 1980) and 38 modern cultivars (released after 2010) using a recently published dataset[11,42]. In short, we analyzed the impact of variations in global radiation (GR), precipitation (PR), and temperature across 91 developmental time windows, ranging from 900 °Cd before to 900 °Cd after heading. This range covers the period from the double-ridge stage to grain desiccation. To quantify these effects, we assessed the mean environmental variables within 100 °Cd time windows and evaluated their impact on yield and yield components for each cultivar by comparing models with and without the influence of a given environmental variable. The coefficient of the environmental effects characterizes the positive or negative effects of an environmental variable on yield components in a given time-window[11].

## Reporting summary
Further information on research design is available in the Nature Portfolio Reporting Summary linked to this article.

## Data availability
The complete sets of BLUEs values used in this study have been deposited in the Zenodo repository: [https://doi.org/10.5281/zenodo.17723507][107].

## Code availability
No custom algorithm was used in this study. All code used to analyse the data are available at [https://doi.org/10.5281/zenodo.17723507][107].

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

## Acknowledgements

Funding for this study was provided by Deutsche Forschungsgemeinschaft (German Research Foundation, DFG). T.-W.C. was funded under "Emmy Noether Programm", project number 442020478. T.-W.C., R.J.S., A.S., and E.H. were funded by DFG under project numbers 518863370, 518783157, 518913298 and 518914346, respectively. T.-W.C. was funded under project number 545869818 to analyse the stage-specific responses. We thank Prof. Dr. Hartmut Stützel and Dr. Carolin Lichthardt to provide the original data published in the dissertation Lichthardt (2020) for our analyses.

## Author contributions

T.-W.C. conceived the data synthesis with input from R.J.S., A.S., E.H., and B.W. T.-C.W. analyzed the data with inputs from M.M., E.V.A., B.A., E.G., L.F., and A.M. A.M. and E.H. conducted the analyses of selection signals. T.-W.C and T.-C.W drafted the manuscript with input from R.J.S., A.S., M.M., E.V.A., B.A., E.G., and L.F. All authors helped to revise the manuscript.

## Funding

## Competing interests

The authors declare no competing interests.
