## [Peer Review file · Nature Communications]

Plasticity of source-sink dynamics contributes to wheat yield stability

Corresponding Author: Professor Tsu-Wei Chen

Version 0:

Reviewer comments:

Reviewer #1

(Remarks to the Author)

Paper entitled "Plasticity of source-sink dynamics contributes to wheat yield stability", the study presents a compilation of large-scale datasets describing historical breeding progress for source- and sink- related traits. The initial wiring diagrams concepts provided foundational approach for source- and sink-related traits for wheat improvement. A meta-analysis focusing on 61 traits using a german winter wheat panel of 202 cultivars (released from 1963 to 2018) were the main dataset. The study is quite robust but in overall the narrow based of the dataset focusing on 6 datasets from german winter wheat cultivars limits the broader implications of this study. How relevant in a global scale comparing to other regions of wheat is this study? There are several questions on how comparable are results when derived from different datasets, with different number of genotypes tested, without equal balance of environments explored, and under different experimental conditions – can we truly compare traits outcomes derived from different datasets under different GxExM conditions? It is not entirely clear, how are the authors adjusting and comparing traits, using a relative breeding progress is a potential approach but there is an assumption that traits are additive on their effect to yield. Most of the calculations are based on this approach and should be more properly documented, the plasticity term is not clearly outlined. The point based on indirect impact on nitrogen economy is poorly supported since a direct measurement of plant N status is not available. Clearly, the manuscript presents interesting results, but I cannot effectively assessed the level of comparability across traits and how results can provide a fair comparison since the data derived from each different trait comes from diverse GxExM combinations even under different experimental settings. I think that this paper is more relevant for Nature Plants or Scientific Data journals, since the scope is narrowly based due to the datasets compiled on this analysis, mainly all german winter wheat varieties.

Reviewer #2

(Remarks to the Author)

Plasticity of source-sink dynamics contributes to wheat yield stability

The paper has an impressive data set and interpretation indicates patterns in terms of how plasticity has enabled yield maximization. This kind of breakthrough is needed to boost yield gains by prioritizing traits, in terms of those that are likely to be complementary and cumulative in their effect on yield, and those that generally trade-off to maximize the former.

I believe that the paper would benefit greatly from a summary -perhaps in the form of a Table or Diagram- showing the more constitutive traits that have the greatest impact on yield and those that have the most plastic response to maximize RUE and harvest index

When using multiple data sets (Table 1) it helps to include temperature and radiation profiles, cycle length, yield and if possible yield gaps to set the context and breadth of the environment.

For example, a recent comprehensive study of global wheat trials shows a linear negative relationship between Tn and yield, especially during grainfilling in the Tn range 8C-22C. Studies in other crops show the same trend. Therefore, since the data here (Lines 380-385) show an opposite response, information about values of Tn should be presented and discussed in this context.

Lines 380-385

Modern cultivars benefit from higher minimum night temperatures by showing increased thousand grain weight (TGW) during early spike development and enhanced grain number per spike during both the green to yellow anther stage (180–140 °Cd 384 before heading) and the anthesis to pre–grain filling stage (140–260 °Cd after heading; 385 Fig. 5).

Figures 1-3 have space to spell out more of the traits as the number of acronyms becomes overwhelming

Figure 5 is complex and would benefit from a brief didactic written guide of how to enter and use it.

Line 353

"Notably, breeding has enhanced plasticity in several adaptive physiological traits, which underpin the plant's ability to cope with unpredictable environmental fluctuations"

Earlier a list of interactions 'that affect all physiological processes under quantitative genetic control' is presented. It may be worth mentioning that PGR signaling and sensitivity have a role in mediating these processes and show genetic variation.

Finally, I think mixing biotic and abiotic factors is too much of an ask of the readers and it was not clear to me if confounding effects of diseases impacted results. I would suggest biotic effects be published in a companion paper or separately.

END

Version 1:

Reviewer comments:

Reviewer #1

(Remarks to the Author)

The authors provided adequate responses to most of my comments and suggestions. However, there are a few points that still need attention: 1) the methods have been revised and now this section has improved, however, the data should be available so the reviewers can check on the information and see if the authors are following the FAIR principle, since their response was "all data and analysis code will be completely published after acceptance of this article, following the FAIR principles" assuming sharing the data once the paper is accepted. This should not be the case. 2) the point related to nitrogen has not been well addressed, the authors do not have nitrogen content in the plant at different points and calculation of N utilization and remobilization, thus, most of the measured variables are very shallow and they do not reflect in any way N efficiency or N use or N content in the plant - they should avoid making a large speculation based on these determinations. 3) the authors provided a response regarding the use of different data sources, but they should know that the meta-analysis does not solve the problem of comparing data from different sources and with different GxExM, I would recommend to include a section in the discussion to provide some statements presenting the main limitations of the data, the comparison and potentially the outcomes presented in the paper (as the authors disclaimed that the paper has some cons, so they should clearly state those). 4) the point related to biotic stress is not even related to the paper and the authors do not have data to make conclusions related to this point, and the overall response of not following the recommendation for the reviewer is not well justified. A statement can be included only saying that this study is not considering this aspect of biotic stressors. The authors should be careful on building speculation from a meta-analysis and only restrict their conclusion based on their data and limited outcomes.

Reviewer #2

(Remarks to the Author)

Thank you for addressing the changes or explaining the reasons if not

Version 2:

Reviewer comments:

Reviewer #1

(Remarks to the Author)

The authors provided adequate responses to my comments and suggestions, and I think that the manuscript is now in a much better shape for publication.

Response to the reviewers for Wang et al. “*Plasticity of source-sink dynamics contributes to wheat yield stability*”

We would first like to express our appreciation to all reviewers for their suggestions, corrections, and highly constructive comments that lead to considerable improvement of the manuscript. In addition, we are grateful to the editors who gave us the necessary time for the revision. We believe that our revised manuscript has addressed all comments and criticisms to our best and made the appropriate changes where necessary in our manuscript. Three major changes have been made in this revision:

- 1) Materials and Methods have been revised to improve clarity about comparability in response to Reviewer 1.
- 2) We rewrote the section “Breeding increases yield resilience under climate change by improving the stage-specific sensitivity of yield components to short-term temperature fluctuations” following Reviewer 2’s comments. In particular, we now specify the temperature ranges used in the regression analyses, which substantially increases the precision of our statements and prevents potential misinterpretation by readers. We revised the corresponding discussion as well.
- 3) We thoroughly re-checked all code used for data analyses and made several minor corrections. Consequently, Figures 1, 2, 3, and S4 have been updated accordingly.

In the following, we attach our detail responses to all reviewers. Their original comments are *italic and blue* and our responses are black. In the main text, we did not use the full track-change mode to maintain the readability, but use a track-change version showing the sections where considerable changes have been made.

Reviewer #1 (Remarks to the Author):

Paper entitled “Plasticity of source-sink dynamics contributes to wheat yield stability”, the study presents a compilation of large-scale datasets describing historical breeding progress for source- and sink- related traits. The initial wiring diagrams concepts provided foundational approach for source- and sink-related traits for wheat improvement. A meta-analysis focusing on 61 traits using a German winter wheat panel of 202 cultivars (released from 1963 to 2018) were the main dataset.

The study is quite robust but in overall the narrow based of the dataset focusing on 6 datasets from German winter wheat cultivars limits the broader implications of this study. How relevant in a global scale comparing to other regions of wheat is this study?

We appreciate the reviewer's recognition of the robustness of our work. We have revised the manuscript to clearly state that the utilized genotype panels represent cultivars which were/are in wide use across central and western Europe, showing the broad implication of our findings beyond Germany. Furthermore, we wish to emphasize that German wheat production and research hold significant global relevance, with average yields in Germany being consistently among the highest in the world (presently 7.4 t/ha compared to the global average of 3.4 t/ha, FAOSTAT 2023). The high yields in central European regions have been attributed to advanced

breeding practices and agronomic management. Therefore, these studies serve as a critical benchmark for advanced breeding and high-input agronomy, making our results highly valuable for regions aiming to sustain, intensify and optimize production worldwide.

There are several questions on how comparable are results when derived from different datasets, with different number of genotypes tested, without equal balance of environments explored, and under different experimental conditions – can we truly compare traits outcomes derived from different datasets under different GxExM conditions?

We agree that interpreting combined results necessitates a careful consideration of the underlying G x E x M context for each trait, and we will detail our responses to your specific, important questions about comparability in the following sections. Before that, we would like to underscore that this study uses meta-analysis precisely because it is nearly impossible in a common experimental setup to measure this number and type of traits in a large number of cultivars with the necessary detail. Our aim was to overcome this logistical challenge and provide a comprehensive overview of the contribution of plasticity in source-sink dynamic traits to yield stability. While non-overlapping genotype datasets are common in meta-analysis, the partially and largely overlapping genotypes and environments across our trials actually provide a useful and robust basis for comparison and the identification of overarching trends.

It is not entirely clear, how are the authors adjusting and comparing traits, using a relative breeding progress is a potential approach but there is an assumption that traits are additive on their effect to yield. Most of the calculations are based on this approach and should be more properly documented, the plasticity term is not clearly outlined.

We thank the Reviewer for pointing out the lack of clarity in our Material and Methods and for raising the question about trait comparability. We have intensively revised the Materials and Methods (L588-612) to significantly improve clarity and to explicitly emphasize the comparability between traits. To ensure maximum transparency and reproducibility and to document our approach properly, all data and analysis code will be completely published after acceptance of this article, following the FAIR principles, consistent with our previous work (Wang et al., 2025, *Journal of Experimental Botany*. In press. <https://doi.org/10.1093/jxb/eraf191>; Wang et al., 2025 *Scientific Data* 12:64; Wang et al., 2023. *Theoretical and Applied Genetics* 136:34.).

We regret the misunderstanding regarding the assumption about additive effects of traits on yield. It is to be clarified here that no such assumption was made, as the interpretation of relative breeding progress depends entirely on the underlying mathematical relationship. For instance, in the case of yield components (where Yield = Biomass x Harvest Index, or Yield = Spike Number x Kernel per Spike x Kernel Weight), the relative breeding progress can be mathematically interpreted in a multiplicative manner and can thus be used to accurately dissect the proportional contribution of different components to the overall breeding gain. This has been discussed in Lichthardt et al. (2020, *Frontiers in Plant Science* 10: 1771).

The point based on indirect impact on nitrogen economy is poorly supported since a direct measurement of plant N status is not available.

In our study, we showed breeding progresses in traits related to nitrogen economy (chlorophyll degradation rate, canopy green duration, nitrogen content in leaves and spikes). This was our idea to mention nitrogen economy in our original text (L199 and L475 in the revised version). To avoid confusion, we replaced “nitrogen economy” by nitrogen uptake or nitrogen use in these sections.

Clearly, the manuscript presents interesting results, but I cannot effectively assessed the level of comparability across traits and how results can provide a fair comparison since the data derived from each different trait comes from diverse GxExM combinations even under different experimental settings.

We are delighted that Reviewer 1 finds our results interesting and we acknowledge the concern regarding the comparability of datasets with diverse GxExM structures. We sincerely hope that our previous responses and the revised sections in Materials and Methods have addressed some of these concerns. We agree that an unbalanced number of genotypes or environments can potentially introduce weighting bias, which is a common challenge inherent in every meta-analysis. To mitigate this, subsets of cultivars were carefully selected before experiments to ensure the representativeness of each panel (detailed in Supplementary Methods). When comparing plasticity, different sub-panels are also analyzed separately (for example, tiller number and leaf morphology; Fig. 3, Fig. S4). Importantly, the primary aim of this meta-analysis was not the direct comparison of individual trials, but rather the identification of overarching, robust trends across the entire data collection. Achieving representative results and maximizing statistical power necessitated the inclusion of datasets with sufficient variation in GxExM combinations, even if some were structurally unbalanced, rather than restricting the analysis to a limited, balanced subset. We sincerely share the concern of the reviewer and believe that we have done our best to balance the pros and cons of different aspects, as we have elucidated our thoughts about comparability in the original manuscript (L101, L152, L259, L301-307, L592-595 and L609-612).

I think that this paper is more relevant for Nature Plants or Scientific Data journals, since the scope is narrowly based due to the datasets compiled on this analysis, mainly all German winter wheat varieties.

We apologize if the scope of our cultivar panel was not sufficiently clear to the reviewer; we hope our previous clarifications have resolved this. To be explicit: the panel includes not only German winter wheat varieties but also the most prevalent and widely cultivated varieties across central Europe from the past 60 years, providing detailed insight into the mechanisms of wheat productivity gains in one of the world’s most important wheat-producing areas. Furthermore, we want to mention that the significance of our work extends beyond the specific data set. The methods, analytical advances, and conceptual frameworks developed in this study are highly relevant and readily transferable to research involving other crops and diverse agricultural regions globally. Germany serves as a representative model system for temperate cereal breeding, and high-quality, deeply curated national datasets are recognized as essential foundations for wider-scale comparative analyses and method validation in global crop breeding research. We believe that our results have clear implications for breeding strategies, phenotyping, and genotype-by-environment analysis, which are core topics for the broad and international audience of *Nature*

Communications. The manuscript was specifically submitted as a contribution to the *Nature Communications* article collection “Plant responses and adaptations to abiotic stress”, for which we believe the content is particularly suited.

Reviewer #2 (Remarks to the Author):

Plasticity of source-sink dynamics contributes to wheat yield stability

The paper has an impressive data set and interpretation indicates patterns in terms of how plasticity has enabled yield maximization. This kind of breakthrough is needed to boost yield gains by prioritizing traits, in terms of those that are likely to be complementary and cumulative in their effect on yield, and those that generally trade-off to maximize the former.

I believe that the paper would benefit greatly from a summary -perhaps in the form of a Table or Diagram- showing the more constitutive traits that have the greatest impact on yield and those that have the most plastic response to maximize RUE and harvest index.

We sincerely thank the reviewer for recognizing the strength of our dataset and the value of our interpretations. As you pointed out, our analyses strongly support the view that considering complementary traits is essential for understanding yield stability.

We have made every effort to summarize our findings in the wiring diagram (Fig. 5). However, further simplifying this network into a rather linear hierarchy that quantifies the “greatest impact” on yield, by plastic response for maximizing RUE, or harvest index, is difficult, and partly contradictory to our message. A key conclusion of our study is that plasticity of traits, and the way breeding history has shaped them, determines a cultivar’s ability to cope with unpredictable and fluctuating environments. Because of this plasticity, the traits exerting the largest effect on yield via RUE and HI are context-dependent rather than universally generalizable.

We hope the reviewer understands this challenge and agrees that, although Fig.5 presents a more complex network, it nevertheless provides a consolidated and accurate synthesis of our findings.

When using multiple data sets (Table 1) it helps to include temperature and radiation profiles, cycle length, yield and if possible yield gaps to set the context and breadth of the environment.

We thank Reviewer 2 for suggesting the inclusion of environmental information in the dataset summary table (Table1). The growing phases and environmental conditions differed considerably between locations, seasons and years. Therefore, adding these details may not provide meaningful or comparable information across datasets. However, we considered this comment as important and integrated mean temperature and global radiation into the Table1 in the revision⁷.

For example, a recent comprehensive study of global wheat trials shows a linear negative relationship between T_n and yield, especially during grain-filling in the T_n range 8C-22C. Studies in other crops show the same trend. Therefore, since the data here (Lines 380-385) show an opposite response, information about values of T_n should be presented and discussed in this

context. “Lines 380-385 Modern cultivars benefit from higher minimum night temperatures by showing increased thousand grain weight (TGW) during early spike development and enhanced grain number per spike during both the green to yellow anther stage (180–140 °Cd before heading) and the anthesis to pre–grain filling stage (140–260 °Cd after heading; Fig. 5).”

We completely agree with this critical comment and we would like to thank the reviewer for pointing this out. We specified the range of temperature that is valid for our findings in Fig. 4 in the revised manuscript (L294-334). Here we show one example in L303-307:

“...we observed a significant positive effect of increased T_{min} on thousand grain weight (TGW) during early spike development (double ridge to terminal spikelet development, 700 -580 °Cd before heading) in both old and modern cultivars (Fig. 4A, 4B; where T_{min} ranged from $0.1\pm 0.5^{\circ}\text{C}$ to $11\pm 0.85^{\circ}\text{C}$)”

In this example, we specified that positive effect of increased T_{min} on thousand grain weight (TGW) during early spike development is valid within the temperature range between 0.1-11.0°C. We add this information in all other results.

Unfortunately, we are not sure which global wheat trials the reviewer is referring. We made several guesses (e.g. Xiong et al., 2024. *Nature Climate Change* **13**:869-875 and Giménez et al., 2025. *Field Crops Research* 110142) but we are really not sure. In our analyses, T_{min} falls into similar range (8-22°C). Although not significant (Fig. 4A, 4C), we do see negative trends of increased temperature to yield in old and new cultivars during grain filling stage (450-550°Cd after heading), probably due to its effect on the number of grains per spike (Fig. 4B, 4D). We discussed this in the revised manuscript (L392-397) and pointed out the differences in experimental setup: prolonged increased temperature treatment in other analyses, vs. the focus on short time windows of 2-3 days in our work.

Figures 1-3 have space to spell out more of the traits as the number of acronyms becomes overwhelming

We thank Reviewer 2 for this suggestion regarding the figure acronyms. While we share the desire to avoid overwhelming the figures with abbreviations, we prioritized maintaining the minimum necessary font size to ensure the readability and comprehensive information transfer of all included traits. Should our manuscript be accepted for publication, we will gladly follow any editorial instructions regarding the scaling or potential consolidation of acronyms and traits in the final figures.

Figure 5 is complex and would benefit from a brief didactic written guide of how to enter and use it.

We thank Reviewer 2 for the comments. We rewrite the legend for Fig. 5 with brief guide about how to read the figure.

Line 353 "Notably, breeding has enhanced plasticity in several adaptive physiological traits, which underpin the plant's ability to cope with unpredictable environmental fluctuations" Earlier a list of

interactions 'that affect all physiological processes under quantitative genetic control' is presented. It may be worth mentioning that PGR signaling and sensitivity have a role in mediating these processes and show genetic variation.

We thank Reviewer 2 for this comment. We adapted our text to include this insight in L52 and L376.

Finally, I think mixing biotic and abiotic factors is too much of an ask of the readers and it was not clear to me if confounding effects of diseases impacted results. I would suggest biotic effects be published in a companion paper or separately.

We thank Reviewer 2 for the suggestion to separate biotic from abiotic results. While this could indeed improve readability, our findings show that resistance to biotic stress contributes substantially to source capacity, thereby supporting sink accumulation and overall yield formation. For example, fungal resistance strongly influences functional green leaf area and thus directly affects source capacity and associated sink processes. We still consider biotic stress an important aspect to be pointed out because crops in the field are simultaneously exposed to biotic and abiotic stresses, and because breeding progress must address these factors jointly rather than in isolation. Therefore, we consider it important not to give the impression that biotic stress can be treated as negligible in our manuscript.

Response to the reviewers for Wang et al. “*Plasticity of source-sink dynamics contributes to wheat yield stability*”

We would first like to express our appreciation to all reviewers for their suggestions, corrections, and highly constructive comments that lead to further improvement of the manuscript. We believe that our revision has addressed all comments and criticisms to our best and made the appropriate changes where necessary in our manuscript.

In the following, we attach our detail responses to all reviewers. Their original comments are *italic and blue* and our responses are black. In the main text, we use a track-change version showing the sections where considerable changes have been made.

Reviewer #1 (Remarks to the Author):

The authors provided adequate responses to most of my comments and suggestions. However, there are a few points that still need attention:

1) the methods have been revised and now this section has improved, however, the data should be available so the reviewers can check on the information and see if the authors are following the FAIR principle, since they response was "all data and analysis code will be completely published after acceptance of this article, following the FAIR principles" assuming sharing the data once the paper is accepted. This should not be the case.

We sincerely thank you for your commitment to open data policies, which we fully share. We have now created a private link to allow the reviewer to access our data prior to acceptance (<https://box.hu-berlin.de/f/6b7d07448ee64d57ba49/>), and we welcome this opportunity to highlight our previous adherence to and engagement with FAIR data principles:

- 1) Wang et al., 2023. *Theoretical and Applied Genetics* 136:34.
In this publication (<https://link.springer.com/article/10.1007/s00122-023-04264-7>), we published the full dataset on Zenodo (<https://doi.org/10.5281/zenodo.4729636>), published the software *toolStability* on Zenodo (<https://doi.org/10.5281/zenodo.5804212>) and CRAN (<https://cran.r-project.org/web/packages/toolStability/index.html>). Furthermore, a repository for reproducing the figures in this publication is available both on on GitHub (https://github.com/llustratien/Wang_2023_TAAG) and Zenodo (<https://doi.org/10.5281/zenodo.7562420>).
- 2) Wang et al., 2025 *Scientific Data* 12:64.
In this publication, all data are deposited and described on Figshare (<https://doi.org/10.6084/m9.figshare.27910269>), including the raw data, metadata and R-script to for removing outliers, combining data sources, generating data visualizations. Further codes to reproduce the results in this publication is publicly available at https://github.com/tillrose/BRIWECS_Data_Publication (pre-processing and visualization) and https://github.com/llustratien/Scientific_Data_Analysis.
- 3) Wang et al., 2025, *Journal of Experimental Botany*. In press.
<https://doi.org/10.1093/jxb/eraf191>. In this publication, we published all data and code as previous two publications here https://github.com/llustratien/JXB_analysis

Here we would like to share the dataset with reviewers via this anonymous private link:

<https://box.hu-berlin.de/d/b83109bbfe994ab18fcc/>

We would be very pleased to incorporate the reviewer's suggestions regarding data documentation and the reproducibility of our results. Maximizing the transparency and reusability of our dataset is also a major priority for us should the manuscript be accepted for publication.

2) the point related to nitrogen has not been well addressed, the authors do not have nitrogen content in the plant at different points and calculation of N utilization and remobilization, thus, most of the measured variables are very shallow and they do not reflect in any way N efficiency or N use or N content in the plant - they should avoid making a large speculation based on these determinations.

We thank Reviewer 1 for their careful attention to terminology and for pointing out the potential misunderstanding regarding nitrogen. We have tried again to revise the manuscript accordingly to clarify this point and avoid any ambiguity as following:

Revision with track-change (L198-203):

~~While direct measurements of nitrogen uptake are unavailable, our results indicate that the nitrogen uptake has been indirectly improved through breeding and explain~~ Also, breeding improved ~~ment in~~ key grain quality traits, such as grain protein content (+13%) and falling number (+4%), alongside yield, indicating coordinated progress in both carbon assimilation and nitrogen acquisition

Revision with track-change (L477-479):

~~Although direct data on nitrogen uptake are lacking, our results indicate that the nitrogen use has been modified during breeding. This is reflected in the~~ Our results indicate improved coordination between canopy stay-green and grain filling, exemplified by a significant reduction in chlorophyll degradation rate and delayed leaf senescence in modern cultivars which enables the plant a longer period of assimilates production.

3) the authors provided a response regarding the use of different data sources, but they should know that the meta-analysis does not solve the problem of comparing data from different sources and with different GxExM, I would recommend to include a section in the discussion to provide some statements presenting the main limitations of the data, the comparison and potentially the outcomes presented in the paper (as the authors disclaimed that the paper has some cons, so they should clearly state those).

We thank the Reviewer's recommendation to include the limitations associated with combining heterogeneous data sources in a single meta-analysis and for suggesting. In the revised manuscript, we include a dedicated paragraph in the Discussion (L512-539) that clarifies the pros and cons of our data and the comparisons performed.

4) the point related to biotic stress is not even related to the paper and the authors do not have data to make conclusions related to this point, and the overall response of not following the recommendation for the reviewer is not well justified. A statement can be included only saying

that this study is not considering this aspect of biotic stressors. The authors should be careful on building speculation from a meta-analysis and only restrict their conclusion based on their data and limited outcomes.

We thank Reviewer 1 for their careful consideration. In this manuscript, we do not give any focus on biotic stressors and therefore do not intend to make specific statements regarding their effects. Our intention is solely to illustrate breeding progress in disease resistance insofar as it contributes to source capacity, thereby supporting sink accumulation and overall yield formation. For example, resistance to fungal diseases strongly influences the maintenance of functional green leaf area and thus directly affects source capacity and associated sink processes. Importantly, we do not wish to imply that biotic stress can be considered negligible.

In response to this comment, we have added a clarifying sentence to the Discussion section (L454-457):

“Although we do not explicitly focus on biotic stressors, improved resistance to fungal diseases helps maintain functional green leaf area and thereby directly affects source capacity and associated sink processes”

Reviewer #2 (Remarks to the Author):

Thank you for addressing the changes or explaining the reasons if not

Thank you for your positive feedback. We appreciate again the reviewer’s careful evaluation in the previous round and are pleased that our responses and revisions satisfactorily addressed the comments raised.